# Association between atherogenic coefficient and depression in US adults: a cross-sectional study with data from National Health and Nutrition Examination Survey 2005–2018

Lu Zhang [ORCID],[1] Jiahui Yin [ORCID],[2] Haiyang Sun,[3] Jiguo Yang,[4] Yuanxiang Liu[5]

LZ, JY and HS contributed equally.

For numbered affiliations see end of article.

**Correspondence to**
Yuanxiang Liu;
lyxlwtg001@163.com and
Professor Jiguo Yang;
jiguoyang0@126.com

## ABSTRACT

**Objective** The pathogenesis of depression is related to immune inflammatory response. Atherogenic coefficient (AC) is an important indicator of lipid abnormalities, which can lead to immune inflammatory responses. However, no study has investigated the relationship between AC and depression in adult Americans. Therefore, we investigated this relationship.

**Design** This study used a cross-sectional design.

**Setting** The National Health and Nutrition Examination Survey (2005–2018) data were used for this study.

**Participants** A total of 32 502 participants aged 20 years or older who had complete information for AC and depression were included in this study.

**Primary and secondary outcome measures** Depressive symptoms were assessed using the nine-item version of the Patient Health Questionnaire (PHQ-9), with a cut-off point of 9/10 indicating likely depression cases. Weighted logistic regression analyses and the smooth curve fittings were performed to explore the association between AC and depression.

**Results** After adjusting for potential confounders, a single unit increase in AC was associated with a 3% increase in the prevalence of depression (HR=1.03, 95% CI=1.00 to 1.06, p=0.039). The relationship between AC and depression was more obvious in females.

**Conclusions** The AC is positively associated with depression.

## INTRODUCTION

Depression is a clinically common emotional state characterised by persistent sadness or inability to experience pleasure, accompanied by deficits in daily functioning.[1] In 2008, the WHO ranked major depression as the third leading cause of the global disease burden and projected that it would be the number one cause by 2030.[2] More than 300 million people worldwide suffer from major depressive disorder (MDD),[3] affecting about 8% of adults in the USA.[4] Depression can cause various adverse events, seriously endangering lives and global health.[5 6] Current

## STRENGTHS AND LIMITATIONS OF THIS STUDY

⇒ The quality and scale of the National Health and Nutrition Examination Survey database ensured our results' statistical power and reliability.

⇒ A wide range of sociodemographic, lifestyle and physical health covariates were adjusted to reduce residual confounding.

⇒ Its cross-sectional design limited this study, and no causal relationships could be determined.

⇒ The Patient Health Questionnaire (PHQ-9 is a proven, simple and effective tool for identifying the severity of depressive symptoms, but it is not a diagnostic tool for major depressive disorder.

antidepressant treatments are effective, but there are many side effects; for example, antidepressants may increase suicidal thoughts in some people.[7] Evidence supports screening for depression and providing early intervention.[8] Therefore, it is necessary to explore the factors related to depression.

Abnormal lipid metabolism leads to many pathological changes. First, activation of the proinflammatory response leads to a decrease in high-density lipoprotein (HDL) and phospholipids and a compensatory increase in phospholipid-rich low-density lipoprotein (LDL), which in turn slows total cholesterol (TC) metabolism and affects neurotransmitters and neural circuits.[9] However, cytokine signalling in adipose tissue, particularly tumour necrosis factor (TNF), promotes metabolic dysregulation.[10] In addition, some studies have shown that changes in circulating lipid concentrations may be associated with depression.[11] Abnormal lipids are involved in the formation of atherosclerosis. The pathogenesis of atherosclerosis is based on the lipid theory, and the explanation is related to excess cholesterol being the sole cause of lipid deposition in the arterial wall.[12]

Atherosclerosis can cause cardiovascular disease (CVD), stroke, etc., often comorbidities with depression.

The atherogenic coefficient (AC) is an important index for assessing the degree of atherosclerosis, calculated as (TC – HDL)/HDL.[13] Nunes *et al* found that AC was elevated in patients with MDD and bipolar disorder.[14] AC and depression can be controlled using statins and other cardiovascular drugs.[15 16]

Exploring the role of AC in depression may be beneficial for treating depression and its complications. Therefore, we used data from the National Health and Nutrition Examination Survey (NHANES) database to explore the association of AC with depression in adults.

## METHODS

### Study design and participants

Data of the participants in this study were obtained from the NHANES database, a major programme conducted by the Centres for Disease Control and Prevention (CDC) to assess the health and nutritional status of 5000 adults and children in the USA annually.[17] The NHANES database contains demographic, dietary, examination, laboratory and questionnaire data. The National Center for Health Statistics (NCHS) Research Ethics Review Board (ERB) authorised the NHANES study protocols. Further information regarding the NHANES data can be obtained from its official website (http://www.cdc.gov/nchs/nhanes.htm).

Participants in our study were screened according to the following inclusion criteria: (1) aged 20 years or above and (2) participation in laboratory tests on an empty stomach. The exclusion criteria were: (1) incomplete Patient Health Questionnaire-9 (PHQ-9) and (2) no data on TC or HDL cholesterol levels.

### Assessment of depression

The PHQ-9 is used to assess depression. The PHQ-9 contains nine items that capture the frequency of depressive symptoms: appetite problems, fatigue, sleep difficulties, psychomotor retardation or agitation, concentration problems, lack of interest, depressed mood, feelings of worthlessness and suicidal ideation. It is now widely accepted as an accurate and reliable method for screening depression.[18 19] Each question is scored from '0' (not at all) to '3' (nearly every day), with a total score of 0–27, where a score≥10 is considered clinically relevant depression.[20] PHQ-9 sensitivity compared with semistructured diagnostic interviews was greater than previous conventional meta-analyses that combined reference standards. A 10 or above cut-off score maximised the overall sensitivity and specificity for subgroups.[19]

### Assessment of AC

Fasting blood was drawn from individuals aged ≥20 years, and the blood samples were processed, stored and shipped to the Johns Hopkins University Lipoprotein Assay Laboratory at the Ambulator-Testing Center laboratory. HDL cholesterol levels were directly measured in the serum. The apolipoprotein B (apo B)-containing lipoproteins in the specimen were reacted with a blocking reagent that rendered them non-reactive with the enzymatic cholesterol reagent under the assay conditions. Reagents were purchased from Roche/Boehringer-Mannheim Diagnostics (Mannheim, Germany). The method uses sulfated alpha-cyclodextrin in the presence of $Mg^{+2}$, which forms complexes with apoB-containing lipoproteins and polyethylene glycol-coupled cholesteryl esterase and cholesterol oxidase for HDL cholesterol measurement. HDL cholesterol data collected from participants in 2005–2006 were adjusted using the following equation: corrected HDL = (Solomon Park assigned HDL value) × (participant HDL). TC was enzymatically measured in the serum or plasma in a series of coupled reactions that hydrolyse cholesteryl esters and oxidise the 3-OH cholesterol group. All the information can be obtained from https://wwwn.cdc.gov/Nchs/Nhanes/.

### Assessment of covariates

Covariates in this study, including Body Mass Index (BMI), alcohol intake and glycosylated haemoglobin (HbA1c), were used as continuous variables. BMI was measured as weight (kg) divided by height (m) squared with <25.0 kg/m$^2$ indicating normal, 25.0 to <30.0 kg/m$^2$ indicating overweight, ≥30.0 kg/m$^2$ indicating obesity. Alcohol intake was determined by extracting the mean alcohol intake from the first and second dietary surveys, considering a single day's intake for participants who consumed alcohol at least once. Physical activity was self-reported by participants as either inactive, moderate or vigorous. Categorical variables included age (20–40 years, 40–60 years and ≥60 years), sex (male or female) and race/ethnicity (non-Hispanic white, non-Hispanic black, Mexican American, other Hispanic or other race/multiple races). The poverty-income ratio (PIR) was defined as the ratio of family income to poverty threshold (<1 indicating an income below the poverty threshold and ≥1 indicating an income above the poverty threshold. The latter category was further classified into two groups: 1.00 to <2.00 and ≥2.00). Education level was categorised as high school not completed, high school completed or high school graduate and some college or associated degrees pursued. Marital status was defined as married/living with a partner or widowed/divorced/separated/never married. Hypertension (HTN) (defined as systolic blood pressure≥140 mm Hg or diastolic blood pressure≥90 mm Hg) was determined using three blood pressure measurements at different times, an existing diagnosis or evidence of an existing antihypertensive medication regimen. Diabetes mellitus (DM) was defined as either taking glucose-lowering therapies, HbA1c concentration of ≥6.5%, use of antidiabetic medication, oral glucose tolerance test (OGTT)≥11.1 mmol/L, fasting plasma glucose≥7.0 mmol/L or random blood glucose≥11.1 mmol/L. Smoking status was categorised as non-smokers (smoked<100 cigarettes in a

lifetime), former smoker (not currently smoking but have consumed≥100 cigarettes previously) and current smoker (smoking at least≥100 cigarettes every day or some days).

## Statistical analysis

The main concern was whether AC is associated with depression after adjusting for other factors that may influence depression. Continuous variables are expressed as mean±SD, and categorical variables are expressed as percentages. The weighted $\chi^2$ test was used to compare categorical variables between groups, a one-way analysis of variance was used to compare normally distributed variables between groups and the Kruskal-Wallis H test was used to compare variables with a skewed distribution between groups. Variance inflation factors (VIFs) were used to test multicollinearity. Weighted multivariate logistic regression analysis evaluated the independent association between AC and depression. The participants were categorised into four groups based on AC: <1.9310, 1.9310 to <2.6695, 2.6695 to <3.6430 and ≥3.6430. We used three levels of adjustment: model 1 was adjusted for age, sex and race/ethnicity; model 2 was adjusted for the variables in model 1 plus BMI, PIR, educational level and marital status and model 3 was adjusted for the variables in model 2 plus HTN, DM, alcohol intake, smoking status, physical activity and HbA1c. The imputation of missing data was conducted using the missForest R package. This random forest-based technique is highly computationally efficient for high-dimensional data of categorical and continuous predictors.[21] The missing values are presented in the table (online supplemental table S1).

All analyses were performed using R software (The R Foundation, Vienna, Austria) and Empower (X&Y Solutions, Boston, MA, USA). Statistical significance was defined as a two-sided p-value<0.05.

## PATIENT AND PUBLIC INVOLVEMENT

None.

## RESULTS

### Participant characteristics

In this study, 32 502 participants were included (figure 1). Table 1 shows the characteristics of the participants according to their AC. There were statistically significant differences in age, sex, educational level, race/ethnicity, marital status, PIR, alcohol intake, smoking status, physical activity, BMI, HTN, DM, HbA1c, TC and HDL cholesterol between the different AC groups (p<0.05).

In addition to that, among those aged 20 years or older (n=30 441), 3891 (9.79%) participants had missing AC values. The proportion of missing values for the different age groups is shown in online supplemental table S2. Fewer proportions of people between 40 and 69 years old had missing AC values compared with those under 40 and those over 70 years.

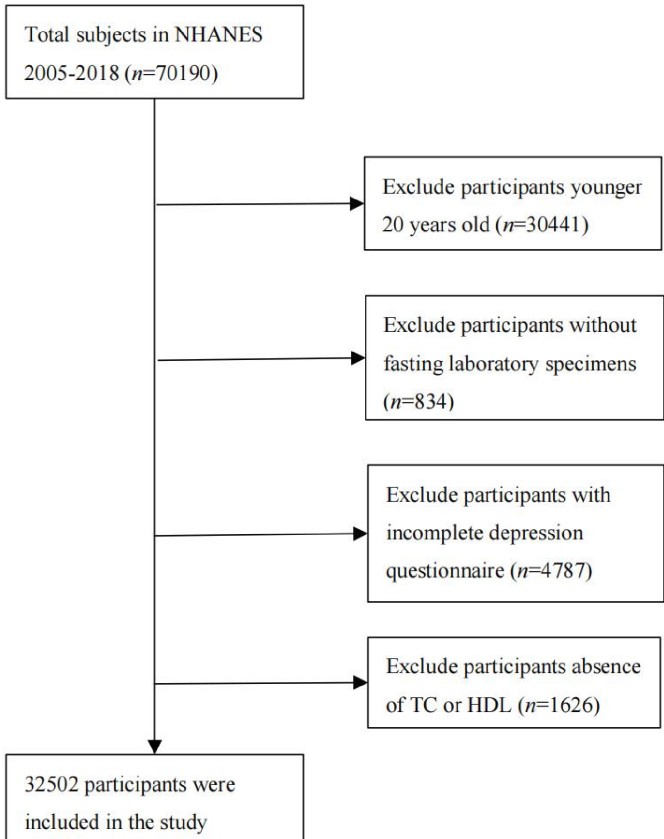

**Figure 1** Flowchart for inclusion of study participants. HDL, high-density lipoprotein; NHANES, National Health and Nutrition Examination Survey; TC, total cholesterol.

We conducted a threshold saturation effect analysis on the data, and the results suggested a linear correlation between AC and depression (log-likelihood ratio=0.051). The results of the threshold saturation effect are displayed in online supplemental table S3. Covariance is generally indicated if the tolerance (Tol) is less than 0.1 or the VIF is greater than 10. Therefore, our results can initially ignore the problem of multicollinearity (online supplemental table S4).

Participants in the lowest AC in Q1 (<1.9310) were likely to be female, younger, more educated, married or cohabitating, non-Hispanic White, wealthier, less physically active, smoked less, consumed more alcohol, had no DM or HTN, higher HDL cholesterol levels, lower BMI, lower HbA1c levels and TC levels.

In contrast, participants with the highest AC in Q4 (>3.6430) were likely to be male, middle-aged, more highly educated, non-Hispanic White, married or cohabitating, wealthier, consumed less alcohol, never smoked, inactive and obese, had HTN, lower HDL cholesterol levels, higher BMI, higher HbA1c and TC levels.

### Association between AC and depression

In the fully adjusted model, we observed a linear relationship between AC and depression (figure 2). The results of the weighted multivariate logistic regression analysis are presented in table 2. AC was positively correlated

**Table 1** Characteristics of the study population, using National Health and Nutrition Examination Survey data from 2005 to 2018 (n=32 502), weighted

| Characteristic | Overall | Atherogenic coefficient quartiles | | | | P value |
| --- | --- | --- | --- | --- | --- | --- |
| | | Q1 | Q2 | Q3 | Q4 | |
| | | (<1.9310) | (1.9310 to <2.6695) | (2.6695 to <3.6430) | (≥3.6430) | |
| Sex (%) | | | | | | <0.001 |
| Male | 48.78 (48.21, 49.35) | 32.75 (31.33, 34.21) | 42.40 (40.95, 43.87) | 54.04 (52.56, 55.52) | 66.36 (65.00, 67.70) | |
| Female | 51.22 (50.65, 51.79) | 67.25 (65.79, 68.67) | 57.60 (56.13, 59.05) | 45.96 (44.48, 47.44) | 33.64 (32.30, 35.00) | |
| Age (%) | | | | | | <0.001 |
| 20 to <40 | 35.93 (34.76, 37.12) | 40.78 (38.93, 42.66) | 36.40 (34.73, 38.10) | 33.15 (31.58, 34.76) | 33.30 (31.69, 34.95) | |
| 40 to <60 | 37.75 (36.83, 38.67) | 29.54 (27.96, 31.18) | 35.21 (33.59, 36.87) | 40.86 (39.37, 42.37) | 45.58 (43.89, 47.29) | |
| ≥60 | 26.32 (25.26, 27.40) | 29.67 (28.04, 31.36) | 28.39 (26.74, 30.09) | 25.99 (24.53, 27.50) | 21.12 (19.78, 22.52) | |
| Educational level (%) | | | | | | <0.001 |
| <High school | 15.44 (14.36, 16.58) | 12.40 (11.24, 13.65) | 14.04 (12.70, 15.49) | 16.18 (14.78, 17.67) | 19.24 (17.90, 20.65) | |
| Completed high school | 23.30 (22.35, 24.27) | 20.49 (19.12, 21.93) | 22.16 (20.82, 23.56) | 24.87 (23.59, 26.20) | 25.75 (24.17, 27.40) | |
| >High school | 61.26 (59.57, 62.92) | 67.11 (65.08, 69.08) | 63.80 (61.69, 65.86) | 58.95 (56.99, 60.89) | 55.02 (52.92, 57.09) | |
| Race/ethnicity (%) | | | | | | <0.001 |
| Non-Hispanic White | 68.41 (65.86, 70.86) | 69.02 (66.42, 71.51) | 68.41 (65.87, 70.84) | 68.23 (65.19, 71.72) | 67.98 (65.01, 70.81) | |
| Non-Hispanic Black | 10.50 (9.27, 11.88) | 13.76 (12.07, 15.64) | 11.32 (9.98, 12.81) | 9.65 (8.42, 11.03) | 7.20 (6.22, 8.32) | |
| Mexican American | 8.46 (7.22, 9.89) | 6.06 (5.10, 7.18) | 7.53 (6.34, 8.93) | 9.50 (8.04, 11.20) | 10.82 (9.06, 12.86) | |
| Other Hispanic | 5.44 (4.65, 6.35) | 4.27 (3.54, 5.13) | 5.22 (4.40, 6.18) | 5.83 (4.87, 6.97) | 6.47 (5.42, 7.70) | |
| Other races/multiple races | 7.18 (6.50, 7.93) | 6.89 (6.08, 7.81) | 7.52 (6.63, 8.51) | 6.79 (5.96, 7.73) | 7.54 (6.67, 8.50) | |
| Marital status (%) | | | | | | <0.001 |
| Married/living with a partner | 64.04 (62.84, 65.22) | 59.25 (57.29, 61.17) | 62.92 (61.31, 64.54) | 66.06 (64.48, 67.61) | 68.04 (66.58, 69.46) | |
| Widowed/divorced/separated/never married | 35.96 (34.78, 37.16) | 40.75 (38.83, 42.71) | 37.08 (35.49, 38.69) | 33.94 (32.39, 35.52) | 31.96 (30.54, 33.42) | |
| PIR (%) | | | | | | <0.001 |
| <1.00 | 13.58 (12.64, 14.57) | 12.69 (11.51, 13.97) | 13.36 (12.18, 14.64) | 12.77 (11.70, 13.92) | 15.51 (14.13, 17.00) | |
| 1.00 to <2.00 | 20.31 (19.33, 21.32) | 18.88 (17.52, 20.31) | 19.48 (18.36, 20.65) | 21.07 (19.63, 22.60) | 21.83 (20.18, 23.57) | |
| ≥2.00 | 66.12 (64.45, 67.74) | 68.44 (66.44, 70.37) | 67.16 (65.18, 69.08) | 66.15 (64.30, 67.96) | 62.66 (60.14, 65.12) | |
| Alcohol intake (g/day) | 9.38 (8.90, 9.87) | 12.33 (11.43, 13.24) | 8.91 (8.22, 9.61) | 7.84 (7.19, 8.49) | 8.43 (7.62, 9.25) | <0.001 |
| Smoking status (%) | | | | | | <0.001 |
| Non-smoker | 54.76 (53.63, 55.88) | 58.23 (56.66, 59.78) | 56.97 (55.26, 58.66) | 54.94 (53.42, 56.45) | 48.79 (47.25, 50.33) | |
| Former smoker | 25.13 (24.28, 26.00) | 24.55 (23.30, 25.85) | 24.60 (23.23, 26.02) | 26.12 (24.65, 27.66) | 25.26 (24.00, 26.56) | |
| Current smoker | 20.11 (19.24, 21.01) | 17.22 (16.03, 18.48) | 18.44 (17.19, 19.75) | 18.94 (17.85, 20.08) | 25.95 (24.45,27.51) | |
| Physical activity (%) | | | | | | <0.001 |

Continued

**Table 1** Continued

| Characteristic | Overall | Atherogenic coefficient quartiles | | | | P value |
|---|---|---|---|---|---|---|
| | | Q1 (<1.9310) | Q2 (1.9310 to <2.6695) | Q3 (2.6695 to <3.6430) | Q4 (≥3.6430) | |
| Inactive | 45.81 (44.18, 47.45) | 39.32 (36.97, 41.72) | 44.08 (41.77, 46.42) | 47.93 (46.10, 49.77) | 52.15 (50.11, 54.17) | |
| Moderate | 28.01 (26.94, 29.11) | 27.48 (25.78, 29.25) | 28.01 (26.30, 29.78) | 29.11 (27.52, 30.76) | 27.43 (25.81, 29.11) | |
| Vigorous | 8.05 (7.55, 8.59) | 8.86 (7.84, 9.99) | 8.74 (7.74, 9.85) | 7.25 (6.40, 8.20) | 7.36 (6.37, 8.48) | |
| Both moderate and vigorous | 18.13 (116.98, 19.34) | 24.34 (22.46, 26.33) | 19.17 (17.54, 20.93) | 15.71 (4.27, 17.25) | 13.07 (11.83, 14.42) | |
| BMI (kg/m²) | 29.10 (28.94, 29.26) | 26.12 (25.94, 26.30) | 28.70 (28.49, 28.90) | 30.28 (30.04, 30.52) | 31.38 (31.17, 31.59) | <0.001 |
| HTN (%) | 38.20 (37.15, 39.25) | 32.90 (31.30, 34.54) | 36.35 (34.76, 37.96) | 40.33 (38.70, 41.98) | 43.34 (41.78, 44.93) | <0.001 |
| DM (%) | 14.38 (13.77, 15.02) | 11.47 (10.59, 12.40) | 13.43 (12.48, 14.45) | 14.90 (13.73, 16.15) | 17.76 (16.69, 18.88) | <0.001 |
| CVD (%) | 8.73 (8.25, 9.22) | 9.42 (8.55, 10.37) | 8.75 (7.97, 9.60) | 8.44 (7.65, 9.30) | 8.28 (7.45, 9.20) | 0.207 |
| Depression (%) | 7.69 (7.24, 8.17) | 6.48 (5.77, 7.26) | 7.16 (6.30, 8.12) | 8.27 (7.46, 9.16) | 8.89 (8.16, 9.68) | <0.001 |
| HbA1c (%) | 5.61 (5.60, 5.63) | 5.44 (5.42, 5.47) | 5.55 (5.53, 5.57) | 5.64 (5.61, 5.67) | 5.82 (5.79, 5.86) | <0.001 |
| TC (mmol/L) | 5.02 (5.00, 5.05) | 4.46 (4.43, 4.50) | 4.77 (4.74, 4.80) | 5.10 (5.07, 5.13) | 5.77 (5.73, 5.80) | <0.001 |
| HDL cholesterol (mmol/L) | 1.38 (1.37, 1.39) | 1.82 (1.80, 1.84) | 1.45 (1.44, 1.46) | 1.24 (1.24, 1.25) | 1.01 (1.00, 1.02) | <0.001 |
| Antidepressants (%) | 13.17 (12.52, 13.84) | 12.65 (11.66, 13.70) | 13.76 (12.67, 14.93) | 13.25 (12.25, 14.33) | 13.01 (11.83, 14.28) | 0.477 |
| Anxiolytics, sedatives and hypnotics (%) | 6.78 (6.33, 7.27) | 7.29 (6.54, 8.12) | 6.78 (6.07, 7.56) | 6.31 (5.56, 7.15) | 6.75 (5.93, 7.68) | 0.339 |

For continuous variables: survey-weighted mean (95% CI), the p-value was by survey-weighted linear regression (svyglm). For categorical variables: survey-weighted percentage (95% CI), the p-value was by survey-weighted $\chi^2$ test (svytable).
BMI, body mass index; CVD, cardiovascular disease; DM, diabetes mellitus; HbA1c, glycosylated hemoglobin; HDL, high-density lipoprotein; HTN, hypertension; PIR, poverty-income ratio (ratio of family income to poverty threshold); TC, total cholesterol.

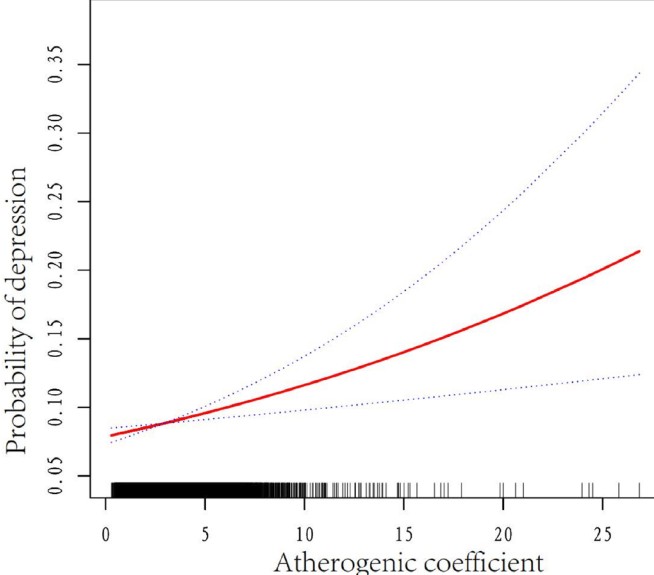

**Figure 2** Association between AC and depression in US adults (n=4759). The black vertical line on the horizontal axis represents the AC distribution, the red line represents the best fit and the difference between the dashed lines represents the 95% CI. The data were adjusted for age, sex, race/ethnicity, body mass index, poverty–income ratio, educational level, marital status, hypertension, diabetes mellitus, alcohol intake, smoking status, physical activity and glycosylated haemoglobin. AC, atherogenic coefficient.

with depression in the crude model (OR)= 1.08, 95% CI: 1.06 to 1.11, p<0.001). A significant association between AC and depression was detected in models 1–3 after adjusting for confounders. In model 3, all variables were adjusted; for every one unit increase in AC, the incidence of depression increased by 3% (OR=1.03, 95% CI=1.00 to 1.06, p=0.039).

After adjusting for age, sex, race/ethnicity, BMI, PIR, educational level, marital status, HTN, DM, alcohol intake, smoking status, physical activity and HbA1c compared

with participants in the first quartile (AC<1.9310), the second group (1.9310 to <2.6695, OR=1.04, 95% CI=0.89 to 1.22, p=0.589), the third group (2.6695 to <3.6430, OR=1.18, 95% CI=1.02 to 1.38, p=0.034) and the fourth group (≥ 3.6430, OR=1.15, 95% CI=0.99 to 1.33, p=0.074) had an increased prevalence of depression (p for trend was significant in all the models).

Furthermore, regarding the interaction between sex and the relationship between AC and depression, the relationship was more significant in females (OR=1.07, 95% CI=1.02 to 1.12, p for interaction=0.027) (table 3).

## DISCUSSION

This cross-sectional study showed an association between AC and depression in adults in the USA. After adjusting for covariates, a positive linear relationship was found between AC and depression.

The details of the mechanism explaining the relationship between AC and depression need to be further explored, and there may be several possible explanations. Lipids and the immune system interact with one another and have a regulatory effect on each other. Dysregulated inflammation promotes susceptibility to depression.[22] Studies have shown that inflammatory cytokines produced in the periphery enter the cells of the central nervous system and can affect neurotransmitters and neural circuits, producing behavioural symptoms of depression.[23] When T lymphocytes are activated, they not only participate in immune inflammation but also directly contribute to the development of depression when functionally impaired.[24 25] Lipid peroxidation and oxidation-specific epitopes are formed, and the levels of antioxidants such as glutathione, glutathione peroxidase, and coenzyme Q10 are reduced, resulting in or aggravating oxidative stress.[26 27]

When lipids are abnormal, the inflammatory, oxidative and nitrosative (IO&NS) pathway is further activated.[28]

**Table 2** Associations of the atherogenic coefficient with depression (n=32 502), weighted.

| | Crude model* | | Model 1† | | Model 2‡ | | Model 3§ | |
|---|---|---|---|---|---|---|---|---|
| | OR (95% CI) | P value | OR (95% CI) | P value | OR (95% CI) | P value | OR (95% CI) | P value |
| Per 1 increase | 1.08 (1.06 to 1.11) | <0.001 | 1.13 (1.10 to 1.16) | <0.001 | 1.07 (1.04 to 1.10) | <0.001 | 1.03 (1.00 to 1.06) | 0.039 |
| Quartiles | | | | | | | | |
| Q1 (AC: <1.9310) | Reference[1] | | Reference[1] | | Reference[1] | | Reference[1] | |
| Q2 (AC: 1.9310 to <2.6695) | 1.11 (0.96 to 1.30) | 0.166 | 1.18 (1.01 to 1.38) | 0.040 | 1.07 (0.91 to 1.25) | 0.431 | 1.04 (0.89 to 1.22) | 0.589 |
| Q3 (AC: 2.6695 to <3.6430) | 1.30 (1.11 to 1.53) | 0.001 | 1.48 (1.26 to 1.74) | <0.001 | 1.24 (1.07 to 1.45) | 0.006 | 1.18 (1.02 to 1.38) | 0.034 |
| Q4 (AC: ≥3.6430) | 1.41 (1.22 to 1.62) | <0.001 | 1.75 (1.50 to 2.03) | <0.001 | 1.32 (1.14 to 1.54) | <0.001 | 1.15 (0.99 to 1.33) | 0.074 |
| P for trend | <0.001 | | <0.001 | | <0.001 | | 0.040 | |

*Crude model
†Model 1: Adjusted for age, sex and race/ethnicity.
‡Model 2: Adjusted for the variables in model 1 plus body mass index, poverty-income ratio, educational level and marital status.
§Model 3: Adjusted for the variables in model 2 plus hypertension, diabetes mellitus, alcohol intake, smoking status, physical activity and glycosylated haemoglobin.
AC, atherogenic coefficient.

**Table 3** Subgroup analysis of the effect of the atherogenic coefficient on depression (n=32 502), weight.

| Subgroup | Number of participants | OR (95% CI) | P for interaction |
|---|---|---|---|
| Sex, n (%) | | | 0.027 |
| Male | 15 954 | 1.05 (1.01 to 1.10) | |
| Female | 16 548 | 1.07 (1.02 to 1.12) | |
| Age, n (%) | | | 0.375 |
| 20–<40 | 10 857 | 1.07 (1.02 to 1.12) | |
| 40–<60 | 10 632 | 1.04 (0.99 to 1.09) | |
| ≥60 | 11 013 | 1.04 (0.97 to 1.11) | |
| Race/ethnicity, n (%) | | | 0.196 |
| Non-Hispanic White | 14 112 | 1.03 (0.99 to 1.08) | |
| Non-Hispanic Black | 6713 | 1.06 (0.98 to 1.15) | |
| Mexican American | 5174 | 1.05 (0.97 to 1.14) | |
| Other Hispanic | 3109 | 1.12 (1.06 to 1.19) | |
| Other race/multiple races | 3394 | 1.10 (0.96 to 1.26) | |
| BMI, kg/m², mean (SD) | | | 0.212 |
| Low | 10 729 | 1.08 (1.00 to 1.17) | |
| Middle | 10 733 | 1.06 (1.01 to 1.12) | |
| High | 10 743 | 1.03 (0.99 to 1.08) | |
| Hypertension, n (%) | | | 0.949 |
| Yes | 13 940 | 1.04 (1.00 to 1.08) | |
| No | 18 562 | 1.06 (1.01 to 1.12) | |
| DM, n (%) | | | 0.670 |
| Yes | 6152 | 1.03 (0.98 to 1.08) | |
| No | 25 761 | 1.07 (1.03 to 1.11) | |

BMI, body mass index (calculated as weight, in kilograms, divided by the square of height, in meters); DM, diabetes mellitus; HTN, hypertension.

At this time, the increase in binding globin and high-sensitivity C-reactive protein (CRP) is accompanied by a significant increase in interleukin-6 (IL-6), TNF-α, interferon-γ, other proinflammatory cytokines and immune inflammation.[29] These factors lead to defects in serotonin and melatonin through the kynurenine pathway, often considered one of the main causes of depression.[30] Activation of the IO&NS pathway leads to mitochondrial and subsequent cellular dysfunction.[31] Previous studies have linked mitochondrial dysfunction in various brain regions to depression.[32]

Statins have also been shown to have antidepressant effects when coprescribed with antidepressants.[33] Lowering the AC index of patients with mood disorders improves CVD outcomes.[34] CVD is a heart and blood vessel disease characterised by myocardial infarction, angina pectoris, heart failure, heart attack and stroke.[35] Longer exposure to depression is associated with significantly increased CVD risk.[36] Factors contributing to the link between depression and cardiac outcome may include alterations in the autonomic nervous system, platelet receptors and function, coagulopathic factors, proinflammatory cytokines, endothelial function, neurohormonal factors and genetic linkages.[37] At the same time, patient compliance with antidepressant treatment is relatively poor.[38] Atherosclerosis is a chronic vascular inflammatory disease associated with oxidative stress and endothelial dysfunction.[39] Atherosclerosis is the underlying cause of CVD, and AC is a major indicator of atherosclerosis.[40] Our results suggest that AC may play a role in depression. AC may indicate the relationship between CVD and depression and a potential target and marker for treating depression or depression combined with CVD. The relevant mechanisms remain to be explored further.

Our study found increased odds of depression with increased AC in adults, demonstrating that controlling AC may be beneficial for preventing depression. Sex may affect this relationship. In the subgroup analysis (table 3), we found a stronger relationship between AC and depression in females. The synergistic effect of oestrogen on cognitive and emotional functions may underlie the association between ovarian hormone fluctuations and depression in females.[41] The induction of indoleamine 2, 3-dioxygenase and deleterious effects of tryptophan catabolising metabolites (TRYCATs) play a role in the pathophysiology of depression. Activation of IDO decreases plasma tryptophan levels and increases TRYCAT synthesis in depressed individuals. Females showed more IDO activation and TRYCAT production after immune challenge than males.[42] Therefore, this sex difference in immune dysregulation may contribute to higher levels of anxiety and depression experienced by females.

This study has some limitations. First, this was a cross-sectional study; therefore, we could not determine a causal relationship between AC and depression. Second, the PHQ-9 is a proven, simple and effective tool for identifying the severity of depressive symptoms, but it is not a diagnostic tool for MDD. Third, the relationship we studied may have been influenced by other confounding factors, which we have not adjusted. Fourth, the differences in demographics and population characteristics in the USA may limit the generalisability of the findings to other countries or regions. Fifth, AC is associated with both depression and CVD, which may potentially affect the nervous system and mental health. At the same time, there are also uncontrollable variables such as lifestyle and regional culture. Finally, although collinearity statistics did not find any significant collinearity, relationships that are beyond statistical p-values may exist.

## CONCLUSIONS

Our research shows that higher AC levels in US adults are positively related to a higher prevalence of depression. Further studies are required to explore the underlying mechanisms and potential benefits of controlling AC levels in patients with depression.

**Author affiliations**
[1]The First Clinical College, Shandong University of Traditional Chinese Medicine, Jinan, Shandong, China
[2]College of Traditional Chinese Medicine, Shandong University of Traditional Chinese Medicine, Jinan, Shandong, China
[3]The Second Clinical Medical College of Guangzhou University of Chinese Medicine, Guangzhou, Guangdong, China
[4]College of Acupuncture and Massage, Shandong University of Traditional Chinese Medicine, Shandong, China
[5]Department of Neurology, Affiliated Hospital of Shandong University of Traditional Chinese Medicine, Jinan, Shandong, China

**Acknowledgements** Thanks to the acupuncture-medicine team. Thanks to Prof. Liu and Prof. Yang for their careful teaching. Thanks to our colleagues for their help. Thanks to all who contributed to the manuscript.

**Contributors** Conceptualisation, LZ and JY; methodology, LZ; software, JY; validation, HS, YL and JY; formal analysis, LZ, JY and HS; investigation, LZ; resources, LZ; data curation, HS; writing—original draft preparation, LZ; writing—review and editing, YL and JY; visualisation, JY; supervision, YL and JY; project administration, LZ. All authors have read and agreed to the published version of the manuscript. The guarantor is LZ.

**Funding** This work was supported by the Shandong Province Traditional Chinese Medicine Science and Technology project key project Z-2023019.

**Competing interests** None declared.

**Patient and public involvement** Patients and/or the public were not involved in the design, or conduct, or reporting or dissemination plans of this research.

**Patient consent for publication** Not required.

**Ethics approval** The National Center for Health Statistics (NCHS) Research Ethics Review Board (ERB) authorized the NHANES study protocols. The data for this study were obtained from the NHANES database.

**Provenance and peer review** Not commissioned; externally peer reviewed.

**Data availability statement** The access policy and procedures are available at https://www.cdc.gov/nchs/nhanes/.

**ORCID iDs**
Lu Zhang http://orcid.org/0000-0002-5932-4566
Jiahui Yin http://orcid.org/0000-0002-4577-1605

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
