## [Reviewer comments · BMJ Open]

ARTICLE DETAILS

TITLE (PROVISIONAL)	Association between atherogenic coefficient and depression in US adults: a cross-sectional study with data from National Health and Nutrition Examination Survey 2005-2018
AUTHORS	Zhang, Lu; Yin, Jiahui; Sun, Haiyang; Yang, Jiguo; Liu, Yuanxiang

VERSION 1 – REVIEW

REVIEWER	Esmaily, Habibollah Mashhad University of Medical Sciences, Biostatistics
REVIEW RETURNED	10-Jun-2023

GENERAL COMMENTS	The paper pretenses the association between CVD and AC index. The manuscript is well organized and written. However, before accepted I have some comments as follows: 1- Update the references.2- Introduction needs to be improved based on your findings.3- I suggest adding SBP and DBP in analysis.4- Do you have any characteristics of drugs for patients in the data? I suggest adding them in Table 1.
--

REVIEWER	Kim, Seon-Jip Seoul National University, Department of Preventive and Social Dentistry, School of Dentistry
REVIEW RETURNED	31-Jul-2023

GENERAL COMMENTS	Currently, there is active research focusing on exploring the connection between cardiovascular and cerebrovascular diseases and mental disorders like depression, using diagnostic indicators. In this context, studies investigating the link between the atherogenic coefficient (AC), which assesses the extent of arteriosclerosis, and depression, have garnered significant interest. Methods 1. Were all data weighted for statistical analyses to account for the complex multistage, stratified, and unequally weighted or clustered sampling design of the NHANES?2. As the author mentioned, the PHQ-9 consists of 9 items used to assess depression. However, in large-scale cross-sectional surveys conducted as self-reports, there is a tendency for depression to be overestimated. To address this, it is crucial to rigorously evaluate depression diagnoses. Previous studies have indicated that a score of 10 or higher is considered indicative of clinical depression, but to provide clearer results, it may be more effective to focus on individuals with scores of 20 to 27 or higher.
---

	These individuals require therapeutic intervention and evaluation by professional institutions, thus ensuring more accurate and meaningful findings. 3. There is typically no universally established criterion for determining the risk levels of the Atherogenic coefficient (AC). The perceived risk associated with AC values can vary among research studies and healthcare professionals. However, AC values below 2 ($AC < 2$) are generally regarded as indicative of a relatively healthy state. On the other hand, AC values of 3 or higher ($AC \geq 3$) are considered indicative of an elevated risk of cardiovascular diseases. While AC is frequently used in clinical practice, it is not a routine test included in standard lipid panels. In some cases, healthcare providers may order AC measurement as an additional assessment, especially when a more comprehensive evaluation of cardiovascular risk is required or when there are specific concerns about a patient's lipid profile. What is the proportion of cases where AC values are missing, and is the percentage of missing values expected to vary with age? Please provide an explanation for this. 4. Investigating the subject's AC value by dividing it into four quartiles is a promising approach. However, it would also be worthwhile to examine whether depression significantly increases beyond a certain AC value. This could involve dichotomizing the participants into a high AC group and assessing the odds ratio for depression in the low AC group to identify any potential significant associations. Results 1. Does Q1 write "McCarron et al. (2021)" in Table 2 mean that it is used as a reference? be clear. 2. Does the first row of Table 2 indicate a trend toward an increase in odds for depression whenever the AC value increases by 1? Be clear. Also describe the statistical methods. 3. For the subgroup analysis (table 3), it seems better to show only the variables that are significant or worth mentioning even if they are not significant, rather than showing all covariates. 4. Since ac is an indicator of arteriosclerosis, cholesterol, glycated hemoglobin, and HDL were also corrected (table 2), but was multicollinearity not considered? Discussion 1. While the hypothesis of an association between AC and depression is well-documented, it is essential to acknowledge the limitations of this study. It would be good to describe the limitations of this study a little more.
--	---

VERSION 1 – AUTHOR RESPONSE

Response to the reviewer 1's comments:

1- Update the references.

Response: Thank you very much for your valuable suggestions. The updated sections in the revised manuscript are marked in red for easy reference.

2- Introduction needs to be improved based on your findings.

Response: We appreciate your constructive comments on our manuscript. We have modified the introduction as follows (Pages 3-4, Lines 43-74):

Depression is a clinically common emotional state characterized by persistent sadness or inability to experience pleasure, accompanied by deficits in daily functioning. In 2008, the World Health Organization (WHO) ranked major depression as the third leading cause of the global disease burden and projected that it would be the number one cause by 2030. More than 300 million people worldwide suffer from major depressive disorder (MDD), affecting about 8% of adults in the US. Depression can cause various adverse events, seriously endangering lives and global health. Current antidepressant treatments are effective, but there are many side effects; for example, antidepressants may increase suicidal thoughts in some people. Evidence supports screening for depression and providing early intervention. Therefore, it is necessary to explore the factors related to depression.

Abnormal lipid metabolism leads to many pathological changes. Firstly, activation of the pro-inflammatory response leads to a decrease in high-density lipoprotein (HDL) and phospholipids and a compensatory increase in phospholipid-rich low-density lipoprotein (LDL), which in turn slows total cholesterol (TC) metabolism and affects neurotransmitters and neural circuits. However, cytokine signaling in adipose tissue, particularly tumor necrosis factor (TNF), promotes metabolic dysregulation. In addition, some studies have shown that changes in circulating lipid concentrations may be associated with depression. Abnormal lipids are involved in the formation of atherosclerosis. The pathogenesis of atherosclerosis is based on the lipid theory, and the explanation is related to excess cholesterol being the sole cause of lipid deposition in the arterial wall. Atherosclerosis can cause cardiovascular disease (CVD), stroke, etc., often co-morbidities with depression.

The atherogenic coefficient (AC) is an important index for assessing the degree of atherosclerosis, calculated as $(TC - HDL)/HDL$. Nunes et al. found that AC was elevated in patients with MDD and bipolar disorder. AC and depression can be controlled using statins and other cardiovascular drugs.

Exploring the role of AC in depression may be beneficial for treating depression and its complications. Therefore, we used data from the National Health and Nutrition Examination Survey (NHANES) database to explore the association of AC with depression in adults.

3- I suggest adding SBP and DBP in analysis.

Response: Thank you very much for your comment. We tried to extract the variables of blood pressure and found that the results of blood pressure extraction in the database were not uniform and could not be treated uniformly. If a single variable were included in the study, there would be a large selection bias.

4- Do you have any characteristics of drugs for patients in the data? I suggest adding them in Table 1.

Response: Thank you for your thoughtful advice. We further extracted relevant data and described the characteristics of drugs for patients. The use of antidepressants, anxiolytics, sedatives, and hypnotics was assessed through questionnaires (Page 6, Lines 148-149).

In addition, there were no significant differences in the use of antidepressants, anxiolytics, sedatives, and hypnotics in participants with CVD ($p > 0.05$) (Page 7, Lines 178-180).

Response to the reviewer 2's comments:

Methods

1. Were all data weighted for statistical analyses to account for the complex multistage, stratified, and unequally weighted or clustered sampling design of the NHANES?

Response: We think this is an excellent suggestion. We re-weighted the analysis of the data and revised the results. The results are as follows (Pages 8-10, Lines 190-197; Page 11, Lines 207-213; Pages 12-13, Lines 225-228):

Table 1. Characteristics of the study population, using National Health and Nutrition Examination Survey data from 2005–2018 (N = 32,502), Weighted

Characteristic	Overall	Atherogenic coefficient quartiles†				p-value
		Q1 (< 1.9310)	Q2 (1.9310 to < 2.6695)	Q3 (2.6695 to < 3.6430)	Q4 (≥ 3.6430)	
Sex (%)						<0.001
Male	48.78 (48.21,49.35)	32.75 (31.33, 34.21)	42.40 (40.95, 43.87)	54.04 (52.56,55.52)	66.36 (65.00,67.70)	
Female	51.22 (50.65, 51.79)	67.25 (65.79, 68.67)	57.60 (56.13, 59.05)	45.96 (44.48, 47.44)	33.64 (32.30, 35.00)	
Age (%)						<0.001
20 to < 40	35.93 (34.76, 37.12)	40.78 (38.93, 42.66)	36.40 (34.73, 38.10)	33.15 (31.58, 34.76)	33.30 (31.69, 34.95)	
40 to <60	37.75 (36.83, 38.67)	29.54 (27.96, 31.18)	35.21 (33.59, 36.87)	40.86 (39.37,42.37)	45.58 (43.89, 47.29)	
≥ 60	26.32 (25.26,27.40)	29.67 (28.04, 31.36)	28.39 (26.74, 30.09)	25.99 (24.53, 27.50)	21.12 (19.78,22.52)	
Educational level (%)						<0.001
<High school	15.44 (14.36, 16.58)	12.40 (11.24, 13.65)	14.04 (12.70, 15.49)	16.18 (14.78, 17.67)	19.24 (17.90, 20.65)	
Completed high school	23.30 (22.35, 24.27)	20.49 (19.12, 21.93)	22.16 (20.82, 23.56)	24.87 (23.59,26.20)	25.75 (24.17, 27.40)	
>High school	61.26 (59.57, 62.92)	67.11 (65.08, 69.08)	63.80 (61.69, 65.86)	58.95 (56.99, 60.89)	55.02 (52.92, 57.09)	

Race/ethnicity (%)						<0.001
Non-Hispanic White	68.41 (65.86, 70.86)	69.02 (66.42, 71.51)	68.41 (65.87, 70.84)	68.23 (65.19, 71.72)	67.98 (65.01, 70.81)	
Non-Hispanic Black	10.50 (9.27,11.88)	13.76 (12.07, 15.64)	11.32 (9.98, 12.81)	9.65 (8.42, 11.03)	7.20 (6.22, 8.32)	
Mexican American	8.46 (7.22, 9.89)	6.06 (5.10, 7.18)	7.53 (6.34, 8.93)	9.50 (8.04, 11.20)	10.82 (9.06, 12.86)	
Other Hispanic	5.44 (4.65,6.35)	4.27 (3.54, 5.13)	5.22 (4.40, 6.18)	5.83 (4.87, 6.97)	6.47 (5.42, 7.70)	
Other races/multiple races	7.18 (6.50, 7.93)	6.89 (6.08, 7.81)	7.52 (6.63, 8.51)	6.79 (5.96, 7.73)	7.54 (6.67, 8.50)	
Marital status (%)						<0.001
Married/Living with a partner	64.04 (62.84, 65.22)	59.25 (57.29, 61.17)	62.92 (61.31, 64.54)	66.06 (64.48, 67.61)	68.04 (66.58,69.46)	
Widowed/Divorced/Separated/Never married	35.96 (34.78, 37.16)	40.75 (38.83, 42.71)	37.08 (35.49, 38.69)	33.94 (32.39, 35.52)	31.96 (30.54, 33.42)	
PIR (%)						<0.001
< 1.00	13.58 (12.64, 14.57)	12.69 (11.51, 13.97)	13.36 (12.18, 14.64)	12.77 (11.70, 13.92)	15.51 (14.13, 17.00)	
1.00 to <2.00	20.31 (19.33, 21.32)	18.88 (17.52, 20.31)	19.48 (18.36, 20.65)	21.07 (19.63, 22.60)	21.83 (20.18, 23.57)	
≥2.00	66.12 (64.45, 67.74)	68.44 (66.44, 70.37)	67.16 (65.18, 69.08)	66.15 (64.30, 67.96)	62.66 (60.14, 65.12)	
Alcohol intake (g/day)	9.38 (8.90,9.87)	12.33 (11.43, 13.24)	8.91 (8.22, 9.61)	7.84 (7.19, 8.49)	8.43 (7.62, 9.25)	<0.001
Smoking status (%)						<0.001
Non-smoker	54.76 (53.63, 55.88)	58.23 (56.66, 59.78)	56.97 (55.26, 58.66)	54.94 (53.42, 56.45)	48.79 (47.25, 50.33)	
Former smoker	25.13 (24.28, 26.00)	24.55 (23.30, 25.85)	24.60 (23.23, 26.02)	26.12 (24.65, 27.66)	25.26 (24.00, 26.56)	
Current smoker	20.11 (19.24, 21.01)	17.22 (16.03, 18.48)	18.44 (17.19, 19.75)	18.94 (17.85, 20.08)	25.95 (24.45,27.51)	
Physical activity (%)						<0.001
Inactive	45.81 (44.18, 47.45)	39.32 (36.97, 41.72)	44.08 (41.77, 46.42)	47.93 (46.10, 49.77)	52.15 (50.11, 54.17)	

Moderate	28.01 (26.94, 29.11)	27.48 (25.78, 29.25)	28.01 (26.30, 29.78)	29.11 (27.52, 30.76)	27.43 (25.81, 29.11)	
Vigorous	8.05 (7.55, 8.59)	8.86 (7.84, 9.99)	8.74 (7.74, 9.85)	7.25 (6.40, 8.20)	7.36 (6.37, 8.48)	
Both moderate and vigorous	18.13 (116.98, 19.34)	24.34 (22.46, 26.33)	19.17 (17.54, 20.93)	15.71 (4.27, 17.25)	13.07 (11.83, 14.42)	
BMI (kg/m ²)	29.10 (28.94, 29.26)	26.12 (25.94, 26.30)	28.70 (28.49, 28.90)	30.28 (30.04, 30.52)	31.38 (31.17, 31.59)	<0.001
HTN (%)	38.20 (37.15, 39.25)	32.90 (31.30, 34.54)	36.35 (34.76, 37.96)	40.33 (38.70, 41.98)	43.34 (41.78, 44.93)	<0.001
DM (%)	14.38 (13.77, 15.02)	11.47 (10.59, 12.40)	13.43 (12.48, 14.45)	14.90 (13.73, 16.15)	17.76 (16.69, 18.88)	<0.001
CVD (%)	8.73 (8.25, 9.22)	9.42 (8.55, 10.37)	8.75 (7.97, 9.60)	8.44 (7.65, 9.30)	8.28 (7.45, 9.20)	0.207
Depression (%)	7.69 (7.24, 8.17)	6.48 (5.77, 7.26)	7.16 (6.30, 8.12)	8.27 (7.46, 9.16)	8.89 (8.16, 9.68)	<0.001
HbA1c (%)	5.61 (5.60, 5.63)	5.44 (5.42, 5.47)	5.55 (5.53, 5.57)	5.64 (5.61, 5.67)	5.82 (5.79, 5.86)	<0.001
Cholesterol (mmol/L)	5.02 (5.00, 5.05)	4.46 (4.43, 4.50)	4.77 (4.74, 4.80)	5.10 (5.07, 5.13)	5.77 (5.73, 5.80)	<0.001
HDL cholesterol (mmol/L)	1.38 (1.37, 1.39)	1.82 (1.80, 1.84)	1.45 (1.44, 1.46)	1.24 (1.24, 1.25)	1.01 (1.00, 1.02)	<0.001
Antidepressants (%)	13.17 (12.52, 13.84)	12.65 (11.66, 13.70)	13.76 (12.67, 14.93)	13.25 (12.25, 14.33)	13.01 (11.83, 14.28)	0.477
Anxiolytics, sedatives, and hypnotics (%)	6.78 (6.33, 7.27)	7.29 (6.54, 8.12)	6.78 (6.07, 7.56)	6.31 (5.56, 7.15)	6.75 (5.93, 7.68)	0.339

For continuous variables: survey-weighted mean (95% CI), the *p*-value was by survey-weighted linear regression (svyglm). For categorical variables: survey-weighted percentage (95% CI), the *p*-value was by survey-weighted Chi-square test (svytable).

Abbreviations: DM, diabetes mellitus; CVD: cardiovascular disease; BMI, body mass index; HbA1c, glycosylated hemoglobin; HDL, high-density lipoprotein; HTN, hypertension; PIR, poverty-income ratio (ratio of family income to poverty threshold); SD, standard deviation

Table 2. Associations of the atherogenic coefficient with depression (n = 32,502), Weighted.

Crude Model ^a		Model 1 ^b		Model 2 ^c		Model 3 ^d	
OR (95% CI)	p -value	OR (95% CI)	p -value	OR (95% CI)	p -value	OR (95% CI)	p -value

Per 1 increase	1.08 (1.06,1.11)	<0.001	1.13 (1.10,1.16)	<0.001	1.07 (1.04,1.10)	<0.001	1.03 (1.00,1.06)	0.039
Quartiles								
Q1 (AC: < 1.9310)	Reference [1]		Reference [1]		Reference [1]		Reference [1]	
Q2 (AC: 1.9310 to < 2.6695)	1.11 (0.96,1.30)	0.166	1.18 (1.01,1.38)	0.040	1.07 (0.91,1.25)	0.431	1.04 (0.89,1.22)	0.589
Q3 (AC: 2.6695 to < 3.6430)	1.30 (1.11,1.53)	0.001	1.48 (1.26,1.74)	<0.001	1.24 (1.07,1.45)	0.006	1.18 (1.02,1.38)	0.034
Q4 (AC: ≥ 3.6430)	1.41 (1.22,1.62)	<0.001	1.75 (1.50,2.03)	<0.001	1.32 (1.14,1.54)	<0.001	1.15 (0.99,1.33)	0.074
p for trend	<0.001		<0.001		<0.001		0.040	

Abbreviations: AC, atherogenic coefficient.

aModel 1: Adjusted for age, sex, and race/ethnicity.

bModel 2: Adjusted for the variables in Model 1 plus body mass index, poverty-income ratio, educational level, and marital status.

cModel 3: Adjusted for the variables in Model 2 plus hypertension, diabetes mellitus, alcohol intake, smoking status, physical activity, and glycosylated hemoglobin.

Table 3. Subgroup analysis of the effect of the atherogenic coefficient on depression (n = 32502), Weight.

Subgroup	Number of participants	OR (95% CI)	P for interaction
Sex, n (%)			0.027
Male	15954	1.05 (1.01, 1.10)	
Female	16548	1.07 (1.02, 1.12)	
Age, n (%)			0.375
20 – < 40	10857	1.07 (1.02, 1.12)	
40 – < 60	10632	1.04 (0.99, 1.09)	

≥ 60	11013	1.04 (0.97, 1.11)	
Race/ethnicity, n (%)			0.196
Non-Hispanic White	14112	1.03 (0.99, 1.08)	
Non-Hispanic Black	6713	1.06 (0.98, 1.15)	
Mexican American	5174	1.05 (0.97, 1.14)	
Other Hispanic	3109	1.12 (1.06, 1.19)	
Other race/multiple races	3394	1.10 (0.96, 1.26)	
BMI, kg/m ² , mean (SD)			0.212
Low	10729	1.08 (1.00, 1.17)	
Middle	10733	1.06 (1.01, 1.12)	
High	10743	1.03 (0.99, 1.08)	
HTN, n (%)			0.949
Yes	13940	1.04 (1.00, 1.08)	
No	18562	1.06 (1.01, 1.12)	
DM, n (%)			0.670
Yes	6152	1.03 (0.98, 1.08)	
No	25761	1.07 (1.03, 1.11)	

Abbreviations: BMI, body mass index (calculated as weight, in kilograms, divided by the square of height, in meters); DM, diabetes mellitus; HTN, hypertension

2. As the author mentioned, the PHQ-9 consists of 9 items used to assess depression. However, in large-scale cross-sectional surveys conducted as self-reports, there is a tendency for depression to be overestimated. To address this, it is crucial to rigorously evaluate depression diagnoses. Previous studies have indicated that a score of 10 or higher is considered indicative of clinical depression, but to provide clearer results, it may be more effective to focus on individuals with scores of 20 to 27 or

higher. These individuals require therapeutic intervention and evaluation by professional institutions, thus ensuring more accurate and meaningful findings.

Response: We greatly appreciate your time and effort in reviewing our manuscript. The PHQ-9 contains nine items that capture the frequency of depressive symptoms: appetite problems, fatigue, sleep difficulties, psychomotor retardation or agitation, concentration problems, lack of interest, depressed mood, feelings of worthlessness, and suicidal ideation. It is now widely accepted as an accurate and reliable method for screening depression^[1-3] (Page 5, Lines 96-100). Depression status was dichotomized based on a PHQ-9 score ≥ 10 . This cutoff point has been shown to have a sensitivity of 88% and a specificity of 88% for detecting major depression^[4].

References

- [1] Kroenke K, Spitzer RL, Williams JB. The PHQ-9: validity of a brief depression severity measure. *J Gen Intern Med.* 2001;16:606-13.
- [2] Lamers F, Jonkers CC, Bosma H, et al. Summed score of the Patient Health Questionnaire-9 was a reliable and valid method for depression screening in chronically ill elderly patients. *J Clin Epidemiol.* 2008;61:679-87.
- [3] Leavens A, Patten SB, Hudson M, et al. Influence of somatic symptoms on Patient Health Questionnaire-9 depression scores among patients with systemic sclerosis compared to a healthy general population sample. *Arthritis Care Res (Hoboken).* 2012;64:1195-201.
- [4] Kroenke K, Spitzer RL, Williams JB, et al. The Patient Health Questionnaire Somatic, Anxiety, and Depressive Symptom Scales: a systematic review. *Gen Hosp Psychiatry.* 2010;32:345-59.

3. There is typically no universally established criterion for determining the risk levels of the Atherogenic coefficient (AC). The perceived risk associated with AC values can vary among research studies and healthcare professionals. However, AC values below 2 ($AC < 2$) are generally regarded as indicative of a relatively healthy state. On the other hand, AC values of 3 or higher ($AC \geq 3$) are considered indicative of an elevated risk of cardiovascular diseases.

While AC is frequently used in clinical practice, it is not a routine test included in standard lipid panels. In some cases, healthcare providers may order AC measurement as an additional assessment, especially when a more comprehensive evaluation of cardiovascular risk is required or when there are specific concerns about a patient's lipid profile.

What is the proportion of cases where AC values are missing, and is the percentage of missing values expected to vary with age? Please provide an explanation for this.

Response: Thank you for your thoughtful comment. In the NHANES database, 39749 adults were included from 2005 to 2018, of whom 3891 (9.79%) had missing AC values. The proportion of missing values for different age groups is shown in Table S2. Fewer proportions of people 40 to 69 years old had missing AC values compared to those under 40 and over 70 years, indicating a greater and more comprehensive concern for disease risk in this age group.

Table S2 Absence of AC in adults

Age (y)	Missing number (n)	Total number (n)	Proportion (%)
20-29	729	6029	12.09
30-39	665	6044	11.00

40-49	527	6060	8.70
50-59	524	5691	9.21
60-69	575	5974	9.63
70-79	425	3730	11.39
≥80	446	2330	19.14

4. Investigating the subject's AC value by dividing it into four quartiles is a promising approach. However, it would also be worthwhile to examine whether depression significantly increases beyond a certain AC value. This could involve dichotomizing the participants into a high AC group and assessing the odds ratio for depression in the low AC group to identify any potential significant associations.

Response: Thank you very much for your comments. We conducted a threshold saturation effect analysis on the data, and the results suggested a linear correlation between AC and depression (log-likelihood ratio (LLR)=0.051). The results of the threshold saturation effect are displayed in Table S3.

Table S3 Threshold effect analysis for association of AC with depression

Outcomes	Depression
Model 1, β (95%)	
Linear effort model	1.04(1.02,1.07)
Model 2, β (95%)	
Infection point (K)	1.2
<K	0.54 (0.29,1.03)
>K	1.05 (1.02,1.08)
LLR	0.051

Results

Does Q1 write “McCarron et al. (2021)” in Table 2 mean that it is used as a reference?
be clear.

Response: We apologize for making such a mistake. We have changed it to reference in Table 1.

Does the first row of Table 2 indicate a trend toward an increase in odds for depression whenever the AC value increases by 1? Be clear. Also describe the statistical methods.

Response: We are very sorry for the confusion. The first row of Table 2 indicates a trend toward an increase in odds for depression whenever the AC value increases by 1. The relationships between influencing factors were analyzed using multivariate logistic regression analysis, adjusted for all significantly associated patient demographics and comorbidities for the respective model. They were used to identify associations between AC and depression. Odds ratios (OR) were reported with 95% confidence intervals (CI). The level of statistical significance was set at $p < 0.05$.

For the subgroup analysis (table 3), it seems better to show only the variables that are significant or worth mentioning even if they are not significant, rather than showing all covariates.

Response: Thank you for your thoughtful advice. We have deleted the content of Table 3 and retained the significant variables: sex, age, race/ethnicity, BMI, hypertension, and DM.

Since ac is an indicator of arteriosclerosis, cholesterol, glycated hemoglobin, and HDL were also corrected (table 2), but was multicollinearity not considered?

Response: We thank the reviewer for pointing out this issue. We indeed should have applied SPSS tests multicollinearity. The results of the threshold saturation effect are displayed in Table SX. Covariance is generally indicated if the tolerance (Tol) is less than 0.1 or the variance inflation factor (VIF) is greater than 10. Our results can initially ignore the problem of multicollinearity (Table S4).

Table S4. Results of collinearity detection

Model		Unstandardized Coefficients		Standardized Coefficients	t	Significance	Collinearity Statistics	
		B	Standard Error				Tolerance	VIF
1	Constant	-0.067	0.01		-6.668	0		
	AI	0.002	0.002	0.01	1.089	0.276	0.153	6.524
	HDL	-0.002	0.006	-0.003	-0.349	0.727	0.193	5.177
	TC	-0.001	0.002	-0.002	-0.312	0.755	0.269	3.723
	GHB	0.001	0.001	0.004	0.973	0.33	0.936	1.068

Dependent Variable: Depression.

Discussion

1. While the hypothesis of an association between AC and depression is well-documented, it is essential to acknowledge the limitations of this study. It would be good to describe the limitations of this study a little more.

Response: We sincerely appreciate your valuable suggestion. We have supplemented the limitations in the manuscript as follows:

This study has some limitations. First, this was a cross-sectional study; therefore, we could not determine a causal relationship between AC and depression. Second, the PHQ-9 is a proven, simple, and effective tool for identifying the severity of depressive symptoms, but it is not a diagnostic tool for MDD. Third, the relationship we studied may have been influenced by other confounding factors, which we have not adjusted. Fourth, the differences in demographics and population characteristics in the United States may limit the generalizability of the findings to other countries or regions (Pages 15, Lines 284-291)

VERSION 2 – REVIEW

REVIEWER	Kim, Seon-Jip Seoul National University, Department of Preventive and Social Dentistry, School of Dentistry
REVIEW RETURNED	20-Sep-2023

GENERAL COMMENTS	The requested revisions have been made in a reader-friendly manner, and the response is satisfactory. Here are the brief recommendations:  1. Are Tables 4-6 intended for inclusion in supplementary materials to explain them, or have they also been explained in the main text? Or are they meant solely for the reviewers' reference? 2. As mentioned in the Discussion, AC has associations not only with depressive aspects but also with cardiovascular aspects. Therefore, AC may potentially impact overall systemic health, including the nervous system or mental well-being. It is essential to acknowledge that various uncontrollable variables, such as lifestyle and regional culture, exist. While Collinearity Statistics were used to assess multicollinearity between variables, it's important to recognize that there may be relationships beyond statistical p-values. 3. In the R1 version of the file, it's not possible to view the Figures. Additionally, there appear to be discrepancies between the content in the Response_to_reviewers file and what's incorporated into the main text. Please verify this.
---

VERSION 2 – AUTHOR RESPONSE

Response to Reviewer 2’s comments:

1. Are Tables 4-6 intended for inclusion in supplementary materials to explain them, or have they also been explained in the main text? Or are they meant solely for the reviewers' reference?

Response: Thank you very much for your valuable feedback. We had submitted Tables 4-6 in the original revision letter as supplementary materials, and we have discussed them in the manuscript (Pages 7, Lines 180-189).

2. As mentioned in the Discussion, AC has associations not only with depressive aspects but also with cardiovascular aspects. Therefore, AC may potentially impact overall systemic health, including the nervous system or mental well-being. It is essential to acknowledge that various uncontrollable variables, such as lifestyle and regional culture, exist. While Collinearity Statistics were used to assess multicollinearity between variables, it's important to recognize that there may be relationships beyond statistical p-values.

Response: We greatly appreciate your suggestions. We have incorporated your comments in the revised manuscript, as follows:

“Fifth, AC is associated with both depression and CVD, which may potentially affect the nervous system and mental health. At the same time, there are also uncontrollable variables such as lifestyle

and regional culture. Finally, although collinearity statistics did not find any significant collinearity, relationships that are beyond statistical *p*-values may exist.” (Page 13-14, Lines 298-302)

3. In the R1 version of the file, it's not possible to view the Figures. Additionally, there appear to be discrepancies between the content in the Response_to_reviewers file and what's incorporated into the main text. Please verify this.

Response: We agree with your viewpoint. We supplemented the Figures in the Main Document, but it did not meet the requirements of the magazine. The editor returned the Main Document and asked to delete the Figures from it. We uploaded each Figure separately.

We have also cross-checked the content of the original Response_to_reviewers document with that of the manuscript, consolidated and corrected some of the abbreviations, corrected problematic areas, and appropriately annotated the manuscript.

VERSION 3 – REVIEW

REVIEWER	Kim, Seon-Jip Seoul National University, Department of Preventive and Social Dentistry, School of Dentistry
REVIEW RETURNED	04-Oct-2023
GENERAL COMMENTS	I have reviewed the manuscript and am pleased to confirm that the requested revisions have been addressed satisfactorily. The revised manuscript is more constructive and contains clearer content compared to the initial draft, making it easier for readers to understand. It appears that there are no further areas requiring modification.

VERSION 3 – AUTHOR RESPONSE